# Artificial Intelligence for Lameness Detection in Horses—A Preliminary Study

**DOI:** 10.3390/ani12202804

**Published:** 2022-10-17

**Authors:** Ann-Kristin Feuser, Stefan Gesell-May, Tobias Müller, Anna May

**Affiliations:** 1Equine Hospital in Parsdorf, 85599 Vaterstetten, Germany; 2Anirec GmbH, Artificial Intelligence Solutions in Veterinary Medicine, 80539 Munich, Germany; 3Equine Hospital, Ludwig Maximilians University, 85764 Oberschleissheim, Germany

**Keywords:** artificial intelligence, deep learning, pose estimation, lameness, equine

## Abstract

**Simple Summary:**

In the expanding field of artificial intelligence, deep learning and smart-device-technology, a diagnostic software tool was developed, which can help distinguish between lame and sound horses and locate the affected limb. As lameness influences the welfare of horses and is often difficult to detect, this tool can help owners and veterinarians in the process of evaluation. The technology is based on pose estimation, which is already used in human and veterinary science to study movement of limbs or bodies without the need to fix any devices onto the object of interest. In this study, 22 horses with unilateral fore- or hindlimb lameness and a control group of eight sound horses were analysed with the program. Based on the results of the program, it was possible to differentiate between horses with fore- and hindlimb lameness and sound horses. Difficult light settings, such as direct sunlight or darkness, or very even-coloured coats, complicate the precise placement of reference points. The analysis and detection with software-generated movement trajectories using pose estimation is very promising but requires further development.

**Abstract:**

Lameness in horses is a long-known issue influencing the welfare, as well as the use, of a horse. Nevertheless, the detection and classification of lameness mainly occurs on a subjective basis by the owner and the veterinarian. The aim of this study was the development of a lameness detection system based on pose estimation, which permits non-invasive and easily applicable gait analysis. The use of 58 reference points on easily detectable anatomical landmarks offers various possibilities for gait evaluation using a simple setup. For this study, three groups of horses were used: one training group, one analysis group of fore and hindlimb lame horses and a control group of sound horses. The first group was used to train the network; afterwards, horses with and without lameness were evaluated. The results show that forelimb lameness can be detected by visualising the trajectories of the reference points on the head and both forelimbs. In hindlimb lameness, the stifle showed promising results as a reference point, whereas the tuber coxae were deemed unsuitable as a reference point. The study presents a feasible application of pose estimation for lameness detection, but further development using a larger dataset is essential.

## 1. Introduction

Lameness is a term that describes a horse’s change in gait, usually caused by pain or mechanical restriction. There are substantial economic losses attributed to lameness in the equine industry, due to interrupted or truncated sports careers, costs of veterinary services, drugs and additional treatment costs, as well as death [1]. Lameness is one of the most common medical issue in equine veterinary medicine [2], and it can affect any horse at any level of training [3,4].

As undetected lameness poses a significant welfare issue for the affected horse, owners and veterinarians need to be capable of recognising changes of gait as early as possible.

Studies have shown that owners are often unable to recognise lameness in their own horses [5] and that identifying whether the horse experiences musculoskeletal pain resulting in lameness can be very difficult, especially for inexperienced riders [6]. On the clinical side, veterinary experience influences subjective lameness evaluation. Veterinary students and recent graduates often exbibit difficulties in identifying the affected leg [7]. Even amongst experienced veterinarians, there is often a lack of agreement on the affected leg in horses with subtle lameness cases [8,9]. Further limitations to subjective lameness evaluation are the inaccuracy of the human eye and the influence of bias due to the assessment and interpretation of lameness after diagnostic anaesthesia [10,11].

Over the years, many technology-assisted methods have been developed to objectively evaluate gait, movement and lameness in horses. These systems can be divided into two major groups, depending on whether they are based on kinetic or kinematic measuring techniques. Kinetics describes the movement of a rigid body, depending only on the action of forces. In contrast, kinematic analysis characterises the spatio-temporal movement of a rigid body, using time and distance as measurable parameters, without considering the forces [12,13,14].

One of the first kinetic instruments for analysing lameness, which is still used in research and clinical cases [15,16], is the force plate [17]. By recording the ground reaction forces from a lame horse, asymmetrical distribution of body weight on the legs can be measured [18]. Though offering very precise data, lameness analysis with the force plate is expensive, time consuming and only applicable in institutions where this measuring platform is available [12,13,19]. Nevertheless, it is still seen as the gold standard in equine lameness evaluation [20,21]. Other options include a force-measuring horseshoe, which can record ground reaction forces. However, the additional weight and size of the shoe potentially influences the movement of the horse, which reduces its value in lameness evaluation [13,22]. The instrumented treadmill located at the University of Zurich, Switzerland [15,18], offers the possibility to measure the ground reaction forces from several consecutive strides and from all four limbs [21]. Still, horses need to be trained to walk on the treadmill, which is time-consuming. In addition, because of its custom-made, relatively expensive characteristics, the treadmill is not suitable for broad clinical use in the field [13,20,21].

Most of the kinematic lameness evaluation systems can be assigned to one of two groups: optical motion capture (OMC) and inertial measurement unit (IMU). OMC systems use infrared cameras with a recording speed between 100–300 Hz, allowing the collection of a large amount of three-dimensional (3D) coordinate data [21]. Most OMC systems capture data using retro-reflective, spherical markers that are attached to the skin over anatomical locations of interest [12,23,24]. In this setup, an OMC system enables precise recording of 3D movement. However, the cost-intensive nature of the equipment and the time-consuming setup largely limits the use of OMC systems to large clinics and universities [14]. In contrast, the functionality of IMUs is based on gyroscopes and accelerometers [14,20]. Usually both sensors work wirelessly and are attached to certain body segments of a horse, using straps or double-sided tape [25]. The number of sensors and the exact placement differ across IMU systems. While a gyroscope measures the angular velocity around an axis, accelerometers measure the velocity and acceleration along a single axis or multiple axes [13,22]. Even though IMUs are portable, they are still relatively cost-intensive and require a certain level of expertise for data collection, analysis and interpretation. Furthermore, the accumulation of drift errors, which are the sum of all minor measuring errors during one analysis, can influence the results and thereby the outcome of the examination [26].

In the last few years, there has been increasing development of these systems [27,28]. Considering the fact that they require markers or inertial sensors, which need to be fixed onto the object of interest, the studied body parts must be defined beforehand [29]. 

In this study, we attempt to combine pose estimation with lameness evaluation in horses. This offers a new approach that ameliorates some of the disadvantages of other objective lameness detection systems. The use of pose estimation offers a non-invasive way to track and record movements for further analysis. The development and use of pose estimation are based on deep learning. As part of the broad scientific field of artificial intelligence (AI), deep learning creates a neural network of multiple layers which relate to each other. By constantly incorporating new data into the network, it can be trained to recognise patterns in high-dimensional data. The significant difference in comparison to other computer programs is the fact that the filtering criteria of these layers are built autonomously from the algorithm itself, instead of by a software engineer [30].

The aim of this study was to evaluate the usability of pose estimation for detecting and marking specific anatomical reference points, using cell-phone videos of horses being lunged on a circle line. A secondary aim was to determine whether pose estimation can be used to differentiate between sound horses and horses with fore- and hindlimb lameness. We hypothesise that, using reference points on the head and forelimbs, it is possible to distinguish between a forelimb-lame and a non-lame horse. Furthermore, we hypothesize that a differentiation between hindlimb-lame horses and non-lame horses by using the stifle and the tuber coxae as reference points is feasible. 

## 2. Materials and Methods

### 2.1. Technology

#### 2.1.1. Deep Learning

In veterinary science, deep learning is already used in many areas. It offers the possibility to improve behavioural studies, for example of drosophila flies or mice [29], or to aid in developing a pain detection model for stabled horses [31]. Other fields of application are image recognition in radiology, such as the automatic classification of canine thoracic radiographs [32], or in equine ophthalmology, integrated in a diagnostic application with a focus on equine uveitis [33]. 

#### 2.1.2. Pose Estimation

Pose estimation allows for the tracking and recording of the movement of humans, animals, or objects without the need to fix any markers or sensors directly onto the subject of interest [29]. For the study of human poses, several well-described programs such as ArtTrack (Saarbrücken, Germany) or Open-Pose already exist [34,35]. After showing promising results in prior studies with pose estimation on animals, the DeepLabCut (2.2rc3 and 2.2.0.6: https://github.com/DeepLabCut/DeepLabCut/tree/v2.2.0.6; accessed on 10 August 2022) program was used in this study [36]. DeepLabCut is a deep convolutional network based on DeeperCut, which is considered one of the best algorithms for pose estimation. In contrast to other pose estimation tools, such as the MPII Human Pose dataset, with approximately 25,000 datasets, DeepLabCut only requires a relatively small number of 200 training images to train a network [29,37]. The functioning of DeepLabCut is based on two main elements. On the one hand, it uses pre-trained residual neural networks (ResNets), which are trained beforehand on ImageNet (resnet_50: http://download.tensorflow.org/models/resnet_v1_50_2016_08_28.tar.gz; accessed on 10 August 2022), a database that provides images for large-scale object recognition models. On the other hand, it is based on deconvolutional layers, which help to increase the visual information inserted into the network and reach spatial probability densities. After being trained with only a small number of labelled images (~200), the algorithm can predict and mark body parts with accuracy comparable to humans [29].

#### 2.1.3. Reference Point Selection

For the pose estimation, 58 reference points, as listed in Figure 1, were determined. Selection criteria were identifiable anatomical landmarks on the horse, with some of these already used and proven in other lameness detection systems [14,38]. There were four markers on the head, four markers on the neck and trunk, 11 on each forelimb from the shoulder down to the hoof and 14 on each hindlimb between the tubera sacrale and the hooves. Each reference point corresponded to one pixel in one picture.

### 2.2. Collection of Data in Investigated Groups

All horses used in this study were assigned to one of three groups: one training group, one analysis group for lame horses and one analysis group for non-lame horses. Detailed information regarding all three groups is summarised in Table A2. Ethical approval for this study was obtained from the ethics committee of Ludwig Maximilians University, Munich, Germany.

Every horse of the three groups received a full orthopaedic lameness examination [39,40] by an orthopaedic specialist (German specialists for equine medicine), including flexion tests. All horses were examined on hard and soft ground in walk and trot on the straight line and on the circle. Horses with any sign of visible gait asymmetry, a positive flexion test or any pathological results in the lameness examination were excluded.

Lameness results were graded according to the AAEP lameness scale by the American Association of Equine Practitioners on a scale from 1 to 5.

All horses of the training group (*n* = 65) were filmed in various environmental surroundings, which included eight different indoor and 14 different outdoor riding arenas with varying sand and soil surfaces. In order to obtain high recognition probabilities on the labelled reference points, diversity in the coat colour of the horses and environmental backgrounds was necessary. Furthermore, care was taken to film in different weather conditions, such as under sunlight or clouded skies, and during different times of the day to obtain a broad spectrum of different video settings. Horses were recorded in walk and trot from the front, the back (11 s in walk and 7 s in trot, respectively), and from both sides on a straight line (12 s in walk and 7 s in trot, respectively). Horses were also recorded on a circle line with an approximate diameter of 12 m on soft ground (1 min in walk and trot) on both hands.

All horses included in the lame group were privately owned horses presented for lameness examination in the Equine Hospital in Parsdorf, Vaterstetten, Germany. In total, 22 horses were examined and included. Permission for the collection and use of data was obtained from the owners beforehand, and detailed information about the lameness history of the horses was documented. As part of the routine lameness examination in this clinic, the horses were first filmed in walk and trot on both hands for one minute on a 12 m diameter circle on soft ground. After performing flexion tests on concrete and examining gait on firm, as well as on soft, ground, horses were subjected to diagnostic anaesthesia. Depending on the results of the examination and the identified anatomical area, the horses underwent diagnostic imaging (radiographs, ultrasound, computed tomography) and treatment based on the diagnosis. The recorded lameness grades varied from 1 to 4 (AAEP). Horses with a lameness degree ≥ 4/5 were excluded from the study, as well as horses that showed lameness on more than one leg.

The non-lame group represents the reference group and consisted of eight horses. All horses were privately owned by one owner/farm. The horses were filmed in walk and trot on a left (CL) and right (CR) circle line for one minute in each gait. Two additional horses were excluded due to positive flexion tests after lameness had been detected during lunging. All video-recordings were taken with an iPhone 11 (Apple), with the resolution set to 1080 p and 30 fps. 

### 2.3. Training the Artificial Intelligence Tool Using Deep Learning

#### 2.3.1. Data Processing and Training

For training the neural network, 454 still frames from 215 videos of the training group were extracted and the predetermined points of interest (reference points, as defined in Section 2.1.3) were labelled manually. To provide high diversity in the training data, attention was paid to select still frames with different limb positioning combined with varying overlay of limbs. Multiple intermediary trainings were conducted to find a suitable network configuration for the neural network. Additionally, frames with predicted poses that had a significant number of outliers were determined and labelled manually to improve the performance of the network. For the final training set of 454 labelled still frames, the ResNet50 network base architecture was utilised. Five percent of the images were reserved for evaluation during training. These images were used to survey the training status of the algorithm. As this application only had access to a limited amount of training data, the evaluation ratio was left at this default value. All hyperparameters related to the neural network and training process were set to the default values of DeepLabCut. This was to ensure that the neural network in this study was based on the stable results of DeepLabCut, using pre-trained and tested networks [29]. 

Initial tests were conducted using full resolution images (1920 × 1080 pixels) to preserve as much of the details as possible, but stable results could not be achieved. By reducing the resolution of the input images, a significant improvement in training was reached. In the end, a resolution of 768 × 432 pixels, which is 40% of the resolution of the original images, was chosen. This represents a balance of reduced image size without losing too much detail. The latest neural network was trained with 550,000 iterations with a resulting loss of 0.0013 of the training data. This low value indicates that the model fit the training data well. During training, the intention is to reach a preferably low value which must not become zero. This would reveal that the algorithm has learned the data by heart.

However, a comparison of training and evaluation data with respect to error probability showed that there was an average error of 2.6 pixels for training data, compared to as many as 8.22 pixels for evaluation data. Given the resolution of 432 pixels in the vertical axis, this error can make a difference of up to ~1.9% between training and evaluation data. Removing outliers with a likelihood below 60% in the predicted points led to an average error for training data of 2.59 pixels and 6.14 pixels for evaluation data. The small difference in error values for the training data shows its already-high certainty, combined with a distinctly lower certainty on unseen evaluation data. For the setup in this study, the threshold for the exclusion of data was set at a certainty of 60% to obtain high reliability for reference point detection, combined with a low error rate. 

#### 2.3.2. Data Analysis and Measurements and Mathematical Calculations in Trot Videos

For the following analysis, only the trot data were used. Each video included one minute of filming time with an average number of 74 strides per video for Warmbloods and 84 strides for German Riding Ponies. All horses of the second group were subdivided in two categories: A = forelimb-lame, B = hindlimb-lame.

##### Forelimb Lameness

The movement pattern of forelimb lame horses is marked by certain, distinguishable alterations. When trotting, a forelimb-lame horse demonstrates a typical, iterative head nod compared to a sound horse [39,40,41]. In an attempt to shift weight away from the painful leg, a left forelimb-lame horse lowers its head when stepping on the sound right leg and lifts the head up when loading onto the lame left leg [40,41]. Thus, to detect forelimb lameness in this study, the movement of the two forelimbs in comparison with the motion of the head was recorded. Reference points on the forelimbs and the neck were chosen. Reference points 17 (Elbow joint left) and 21 (Carpus left) were used for CL, and 18 (Elbow joint right) and 22 (Carpus right) were used for CR. Reference point 4 (poll) shows the movement of the head during trotting on both circles. To be able to distinguish between the left and right stance phase, points 19 (Os carpi accessorium left), 20 (Os carpi accessorium right), 45 (Tarsus left) and 46 (Tarsus right) were selected. For each horse, the recorded trajectory of the reference points from CL and CR were extracted from the program in csv-files and presented in charts. These data were analysed visually.

##### Hindlimb Lameness

Horses with hindlimb lameness show significant changes in their kinematic pattern [42,43]. In this study, two separate analysis parameters were investigated based on these known changes. 

##### Stifle Reference Point 

Horses with hindlimb lameness often present with a decreased protraction of the lame limb [39,42,43]. To compare the step length of both hindlimbs, the horizontal movement of points 43 (Stifle left) and 44 (Stifle right) on CL and CR was recorded and measured. It was estimated that horses with a hindlimb lameness show a shortened stride on the lame leg and, therefore, show a smaller difference between the measured minima and maxima of the stifle point on the lame side.

##### Tuber coxae reference point

As an approved reference point [41,44], the movement of the tuber coxae along the vertical axis was analysed. Studies have demonstrated that hindlimb-lame horses show an increased vertical displacement of the tuber coxae on the lame side [41,44,45]. Thus, it was estimated that horses with hindlimb lameness show a larger difference between the measured minima and maxima on the affected side.

For each horse, the recorded trajectory of the reference points from the CL and CR were extracted from the program in csv files and transferred into an Excel file (Microsoft Excel, Version 16.63.1). To avoid false results due to inaccurate placement of markers by the program, the maximum 5% (95–100%) and the minimum 5% (0–5%) of the recorded frames were excluded from the analysis. The maxima represent the highest measured values (90–95%) and the minima the lowest measured values (5–10%) of the stifle point and the tuber coxae points. 

For the analysis of the stifle point, ${\bar{Max}}_{St}$ (mean value of the stifle maxima) and ${\bar{Min}}_{St}$ (mean value of the stifle minima) for every horse were calculated for the left and the right circle. The differences represent the length of the horizontal distance along which the stifle point is recorded during trotting on each circle:$$DS_{St}\left(\mathrm{CL})=|{\bar{Max}}_{St}\left(\mathrm{CL}\right)-{\bar{Min}}_{St}\left(\mathrm{CL}\right)\right|$$
$$DS_{St}\left(\mathrm{CR})=|{\bar{Max}}_{St}\left(\mathrm{CR}\right)-{\bar{Min}}_{St}\left(\mathrm{CR}\right)\right|$$

For the analysis of the tuber coxae point, ${\bar{Max}}_{Tcox}$ (mean value of the tuber coxae maxima) and ${\bar{Min}}_{Tcox}$ (mean value of the tuber coxae minima) were calculated for both circles. The differences represent the length of the vertical distance between the highest and lowest tuber coxae values during movement on each circle:$$DT_{Tcox}\left(\mathrm{CL})=|{\bar{Max}}_{Tcox}\left(\mathrm{CL}\right)-{\bar{Min}}_{Tcox}\left(\mathrm{CL}\right)\right|$$
$$DT_{Tcox}\left(\mathrm{CR})=|{\bar{Max}}_{Tcox}\left(\mathrm{CR}\right)-{\bar{Min}}_{Tcox}\left(\mathrm{CR}\right)\right|$$

In the next step the difference for the Stifle as a reference point was calculated to compare the CL and CR:$$D_{St}\;\;=|DS_{St}\left(\mathrm{CL})-\;DS_{St}(\mathrm{CR})\right|$$

The values for the tuber coxae measurements were calculated the same way for comparison of CL and CR: $$D_{Tcox}\;\;=|DT_{Tcox}\left(\mathrm{CL})-\;DT_{Tcox}(\mathrm{CR})\right|$$

Mean values ${\bar{D}}_{St}$ were calculated by summing up the $DS_{St}$ of the individual horses, which should be compared, and dividing them by the number of included horses. 

Mean values ${\bar{D}}_{Tcox}$ were calculated the same way with $DS_{Tcox}$.

#### 2.3.3. Statistical Analysis

Diagnostic test properties based on the AI system in comparison to the clinical assessment (reference) were separately assessed for forelimb lameness, hindlimb lameness using the stifle reference point, and hindlimb lameness using the tuber coxae reference point, using 2 × 2 tables. Estimates for diagnostic sensitivity (SE) were calculated as the proportion of clinically lame horses that were correctly classified based on the AI results. Specificity (SP) was calculated as the proportion of clinically healthy horses that were correctly classified based on the AI results. Accuracy (ACC) was calculated as the proportion of correct (positive + negative) classifications based on the AI results. Positive predictive values (PPV), describing the probability that the AI positive result is correct, and negative predictive values (NPV), describing the probability that the AI negative result is correct, were evaluated. The agreement beyond chance (κappa), a statistical value for quantifying inter-rater reliability, was used in this study to measure agreement between clinical scoring of the horses and classification based on the AI. Kappa scores were calculated on the basis of a 3 × 3-table, including forelimb lameness, hindlimb lameness (only using stifle reference point data) and the non-lame control group. Finally, an overall accuracy (OA) was calculated as the percentage of all correctly classified horses based on the AI results [46].

## 3. Results

Of the 22 horses of the lame group, 13 horses were detected with forelimb lameness and nine horses with hindlimb lameness. The results of their analysis, together with the eight horses of the third group, are presented below.

### 3.1. Forelimb Lameness

In total, seven horses were diagnosed as left-forelimb-lame and six as right-forelimb-lame. The lameness degrees ranged from AAEP 1–2/5 in ten horses and AAEP 3–4/5 in three horses. As shown in Figure 2a), the upward and downward movement (“head nod”) of the poll reference point was visually correlated with the loading of the lame and the non-lame limb, respectively. The non-lame horses did not show any signs of repetitive up-and-down motion of the head, as illustrated in Figure 2b). 

### 3.2. Hindlimb Lameness

The lameness degrees ranged from AAEP 1–2/5 in four horses and AAEP 3–4/5 in five horses. Five horses were lame on the left hindlimb, four horses were lame on the right hindlimb. 

#### 3.2.1. Stifle Reference Point

For every hindlimb-lame and every non-lame horse, the difference $D_{St}\;$ was calculated. Results are presented in Table 1 and Table 2. The median score of all $D_{St}$ of the non-lame group was ${\bar{D}}_{St}\left(\mathrm{non}-\mathrm{lame}\right)=0.55$ To verify detectability of hindlimb lameness with the stifle as reference point, a correlation between the lameness grade and the calculated $D_{St}$ was constructed. After all videos were analysed, horses 2, 4, 7 and 9, were all classified with severe lameness and showed a clear difference in the calculated $D_{St}$ compared to the median ${\bar{D}}_{St}$ of the sound group. For horses 3, 5 and 8, graded with subtle lameness, a smaller difference in the calculated $D_{St}$ compared to the median ${\bar{D}}_{St}$ of the sound group could be illustrated. Therefore, a relation between the degree of lameness and the calculated $D_{St}$ could be shown in all horses, except for horse 1. 

In the control group, with a calculated median ${\bar{D}}_{St}=0.55$, all horses only showed small divergences in the comparison between CL and CR, except horse number 7.

#### 3.2.2. Tuber Coxae Reference Point

For every hindlimb-lame and every non-lame horse, the difference $D_{Tcox}$ was calculated. The results of the calculated $D_{Tcox}$ for every hindlimb-lame horse are presented in Table 3, with the non-lame group in Table 4. The median score of all $D_{Tcox}$ of the control group was ${\bar{D}}_{Tcox}$(non-lame) = 1.30. In three out of nine lame horses (horse 3, 5 and 9), the calculated $D_{Tcox}$ corresponded with the lameness, as a larger difference between the measured minima and maxima on the lame side can be shown. In horses 1, 2, 4, 6, 7 and 8, $D_{Tcox}$ indicated lameness on the contralateral non-lame limb. Comparing the median values of the detected lame, the non-detected lame and the non-lame horses, (${\bar{D}}_{Tcox}$(lame) = 1.21, ${\bar{D}}_{Tcox}$(non-detected lame) = 3.08 and ${\bar{D}}_{Tcox}$(non-lame) = 1.30, respectively); therefore, no correlation between lameness, lameness grade and the absence of lameness could be drawn. 

The mean values for SE, SP, ACC, PPV and NPV according to the analysis of the tuber coxae point of nine hindlimb-lame horses and eight non-lame horses are presented in Table 5. In comparison to the clinical assessment, the classification based on AI calculation was perfect (100% SE and SP) for forelimb lameness, close to 90% for hindlimb lameness when using the stifle reference point, but poor for hindlimb lameness when using the tuber coxae reference point (Table 5). The agreement beyond chance (κappa) was κ = 0.92573. Due to the unreliable results and the inapplicability of tuber coxae as a reference point, it was excluded in this setup. An overall accuracy (OA) of 95.3% could be reached (Table A1). 

## 4. Discussion

In this study, the usability of an AI-based program and its capacity, based on the implementation of pose estimation, to detect specific anatomical landmarks of horses was evaluated. Calculations were made based on these data to differentiate between non-lame and unilateral fore- and hindlimb lame horses. Furthermore, the assessments made based on the program were compared to clinical lameness examination.

We believe that the use of a smartphone application in a real-world, equestrian setting would provide a great advantage to the standard lameness examination. Video analysis is non-invasive, and videos can be obtained at any chosen location with no equipment needed, except for a cell phone camera [29]. The ground surface and training facilities can therefore be those to which the horse is accustomed. This is particularly relevant, as studies have shown adaptations in equine movement and gait when, for example, a treadmill is used [12,47]. Videos obtained using a smartphone are easy to transfer via the internet and can be exchanged with veterinary colleagues all over the globe. Deep learning software is a tool which can help to detect fore- and hindlimb lameness in horses. By applying pose estimation to videos of horses filmed on a circle line and further evaluating the generated data, it is possible to detect lameness without additional hardware.

### 4.1. Forelimb Lameness

With the application of the reference points on the forelimbs and the head, forelimb lameness was detectable in this study. The data revealed head nodding as a result of increased weightbearing on the non-lame limb during stance. By contrast, horses within the non-lame control group did not show any consistent head movement asymmetry in rhythm with the steps onto the right or left forelimbs. A sensitivity and specificity of 100% shows that, by viewing the graphical charts, it is possible to differentiate a forelimb lame from a non-lame horse with this application. The next step will be a further development of the program to classify the extracted parameters of head and limb movement in relation to the stride time. This will allow calculation of the measured values and the collection of more specific data.

### 4.2. Hindlimb Lameness

For analysing hindlimb lameness in this setup, different equine anatomical landmarks on the hindlimbs were considered as reference points. In the pre-evaluation, reference points on the tuber coxae and stifle proved to be the most promising in the detection of hindlimb lameness. The tuber coxae have been used as a reference point in various locomotion studies [41,44,45], while the stifle has not been evaluated previously with portable systems in the horse, as it is not feasible to fix an accelerometer onto this point. To the authors’ knowledge, it has been used as a reference point only in studies with OMC [42,48].

#### 4.2.1. Stifle

In this study, a correlation between the degree of lameness and the calculated $D_{St}$ could be shown in eight out of nine horses. Horse 1 displayed a slight difference between CL and CR, which did not correspond to its lameness grade (3–4). This horse was a dark-brown Warmblood with a very even-coloured coat. As mentioned below, the colour of the horses, especially when showing little or no variance, influences the accuracy of the reference points and, consequently, the results. Horse 7 of the control group was filmed during sunset in an outdoor riding arena and part of the arena was still covered in sunshine. This can affect the quality of the video with the sunbeams causing a glare effect. As mentioned above, the error rate for data evaluation was higher compared to the training data when these effects were present. Given the resolution of 432 pixels in the vertical axis, this error can make a difference of up to ~1.9%. Consequently, the reference points cannot be detected correctly in a few frames per circle, which results in a higher percentage of inaccurate placement. A sensitivity and specificity of almost 90% when using the stifle reference point provides promising results in this first setup. Using more labelled data will help to improve and stabilise the placement of the markers despite disadvantageous light conditions and horses with less well-defined anatomical landmarks.

#### 4.2.2. Tuber Coxae

On the other hand, the tuber coxae point was not suitable for use with videos of horses on a circle line. Comparing the median values between the horses detected as being lame, the horses not detected as being lame and the non-lame horses, no correlation between lameness, lameness grade and the absence of lameness could be drawn. Other studies have shown that left and right tuber coxae should be compared at the same time to detect asymmetry [42,44,49]. As videos of horses on a circle line only show one side of the horse, a direct comparison using this setup was not possible. Furthermore, the large divergence of the calculated values in the control group confirms the fact that the tuber coxae are not suitable as a reference point for this purpose in the given setup.

Depending on the choice of reference points, the AI-based classification showed high to perfect agreement with the clinical assessment. The use of pose estimation reduces some of the limitations that contemporary lameness analysis systems must cope with. The EquiMoves system^®^ (www.equimoves.nl, accessed on 10 August 2022) uses four sensors on the trunk and one sensor on each limb. It detects upper-body movement asymmetries in horses. In comparison with other systems that employ fewer IMU sensors, it is possible to determine stride length and certain limb angles for pro- and retraction and for ad- and abduction [14]. Nonetheless, the sensors must be fixed onto the horse, and the number of reference points is limited compared to the program evaluated in this study. Another IMU system is the Equinosis Q Lameness Locator^®^, (Equinosis LLC, Columbia, MO, USA) which uses two accelerometers on the poll and tuber sacrale to measure the vertical maxima and minima of the head and pelvis during movement. A gyroscope attached to the right forelimb detects the stance phase to differentiate between movements of the left and right sides [25,50]. OMC systems such as QHorse from Qualisys Motion Capture Systems^®^ (Qualysis AB, Motion Capture Systems, Göteborg, Sweden) allow marker fixation on different anatomical landmarks of the horse. With the need for a relatively large space to set up the cameras, evaluation and analysis of horses by this method are limited to large clinics and universities, reducing the flexibility and broad use of this system [18,51]. The use of pose estimation for equine gait analysis offers the possibility to record and analyse the movement of almost unlimited anatomical structures on a horse once the program has been adequately trained. Reference points can be selected before and after recording the horse and videos can be taken anywhere, with only a cellphone camera needed on site.

### 4.3. Limitations

There are some limitations in this study. Sample sizes were small, and larger studies on a broader range of patients are needed to derive robust estimates for SE and SP. To this point, a differentiation of the anatomical origin of lameness is not possible due to small study groups and a limited amount of data. With improvement and advanced training of the program, further studies on the comparison of different causes of lameness are planned.

Using this software on a smartphone device, filming must be standardised, as multiple factors can affect the quality of the videos. As mentioned before, bright sunlight and shade lower the quality of the videos. This problem has also been discussed in other studies [29]. Consequently, the DeepLabCut software has been trained to learn how to robustly extract body parts, even with a cluttered and varying background, inhomogeneous illumination, or camera distortion [36]. In our study, evening light or bright sunshine made filming more difficult, and the analysed data became more imprecise. To evaluate the performance of the tool with videos that were not taken under perfect conditions, different light settings were considered. The horses were filmed inside equestrian arenas with windows and other light sources in different locations, as well as in outside riding arenas with different backgrounds (trees, fields, grass, traffic). Nonetheless, the diversity of videos used to train the AI system needs to be increased.

To find the most suitable filming position, 215 videos were evaluated. It showed that filming the horse, trotting on a straight line, from in front, behind, or from the side, did not offer enough steps for evaluation. However, videos filmed from the inner circle provided good consistency and a sufficient number of strides for analysis. In a complete lameness examination, horses should be evaluated on a straight line and on a circle line [39]. There are differences in motion of the torso and the pelvic area when horses’ motions on a straight line and on a circle line are compared [39,52]. With further development and improvement of the program, it should be possible to analyse shorter video sequences on a straight line.

Irregular movements (horses shaking their heads, vocalising or becoming distracted and showing horizontal or vertical head movements) or other horses in the vicinity decreased correct positioning of reference points by the program. This effect did not have much impact on the results, as the chosen videos of horses on a circle line provided sufficient data to evaluate the lameness, despite data outliers.

When the coat or hoof colour of the horse resembled the background, the sand or the ground, it was difficult to recognise the anatomical markers and their locations became imprecise, so they could not be used. The anatomical structures were less prominent in horses that were completely black or white, especially when they were filmed in direct sunlight, so that labelling became demanding or even impossible in some cases, and they had to be excluded from the study. Apart from these rare cases, coat colour did not cause any selection bias; there was variation of colour in all three categories and a large colour spectrum was covered in non-lame and lame horses. The error rate increased when horses were over-weight or had a long winter coat that made anatomical structures less visible. By excluding the maximum and minimum 5% of the measured values, these small errors could be removed from the data. While the reference points were difficult to evaluate under the above circumstances, markers on the “edge” of the horse, as well as on easily visible anatomical structures, such as the nostril, eye or coronary band, were reproducible.

Another limitation was the quality of footing. Deep sand was unstable, causing horses to stumble or show irregular movements that could resemble lameness. This complicates any lameness examination and is not unique to this study. This needs to be considered with regard to the future use of the tool when videos taken by owners or inexperienced veterinarians will be used. As the volume of labelled data grows, the reliability of the program is expected to increase.

Evaluation of error values for training data showed that excluding outliers with a certainty below 60% only reduced the average error from 2.6 pixels down to 2.59 pixels, indicating that it is unlikely to improve with more training on the current model with the same data. It also shows that the network has high uncertainty on unseen evaluation data, which could be solved by having a greater variety of labelled images in the dataset. With additional augmentation through modification of the images, for example, by adding noise or changing colours or brightness, stability in difficult situations could be improved. Additionally, with more data and different hyperparameters this error can be reduced in future iterations of the neural network.

### 4.4. Outlook for the Future

Pose estimation has the potential to improve gait analysis and lameness diagnostics in equine medicine and veterinary science. It can be applied to various gait or training assessments and can be used in various species such as horses, dogs, cats and dairy cattle. Studies have shown that dairy farmers do not recognise lameness in their cattle, even though it has a large impact on animal welfare, milk yield and, therefore, emerging costs [53,54]. With the help of this new, easily applicable pose estimation program, objective lameness evaluation can be efficiently executed, offering various possibilities for veterinary students and veterinarians to improve their abilities to assess horses’ movements and, therefore, improve welfare for the affected animals [31,55].

Studies have shown that the quality of lameness examination improves with years of work experience, as veterinarians expand their skills and become better in detecting lameness [7]. In addition to these years of training, this tool may serve as a valuable system to improve learning quality and to refine and improve the veterinarian’s ability to evaluate equine gait. Experienced veterinarians can use it for confirmation during daily clinical work and to keep records for retrospective evaluation of treatment. With increasingly more data being assessed and used to train the pose estimation tool, it may be possible to detect subtle gait changes, such as mild lameness or ataxia. Another possible use for the tool could be to compare different trainers or training methods. For example, gait analysis using all reference points to show swinging back movements or different swing-phase trajectories could be quantified to assess training efficacy.

## 5. Conclusions

This study demonstrated the feasibility of obtaining accurate measurements and data that match the clinical presentation in moderately lame horses (grade 3–4/5 AAEP). For horses that were only slightly lame (grade 1–2/5 AAEP), the smartphone app provided less distinct measurements, a sign that the program needs more labelled data and training to become more accurate and reliable. Furthermore, extended studies on the feasibility of the different reference points must be obtained, but these preliminary results are regarded as promising with regard to proof of concept.

## Figures and Tables

**Figure 1 animals-12-02804-f001:**
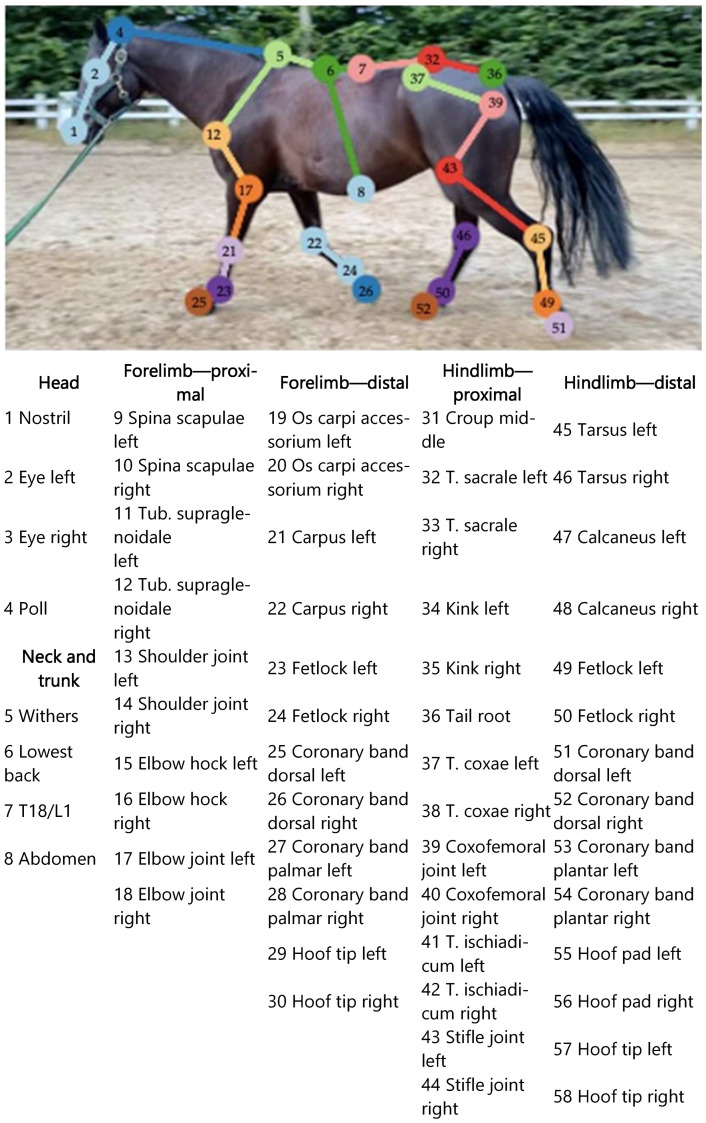
Reference points. Different combinations of reference points can be chosen in the program and offer multiple variations for gait analysis; the picture only shows a selection of the reference points which are enlarged in the image for better visibility. In the program, one reference point corresponds to one pixel. The accurate anatomical locations corresponding to the reference points of the program are listed in Table A1.

**Figure 2 animals-12-02804-f002:**
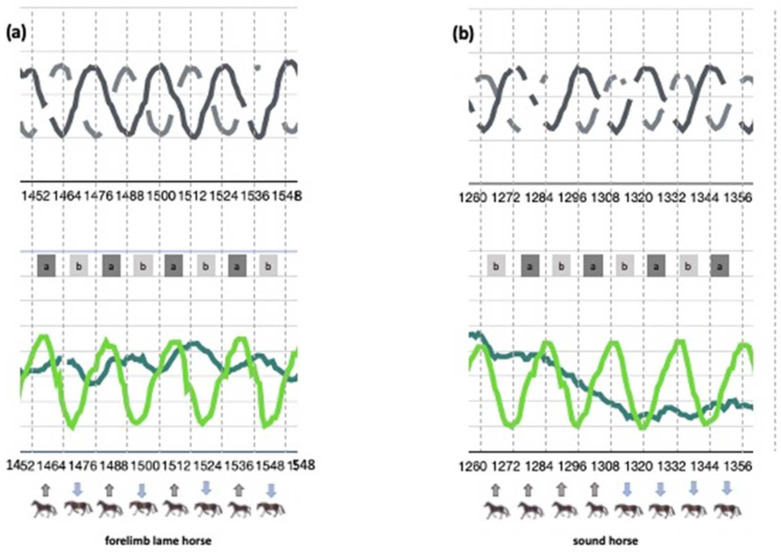
Graphical presentation of forelimb lameness in one representative horse (no. 11) (**a**) compared to a representative non-lame horse (no. 7), (**b**) on a left circle. a in the square = stance phase left forelimb, b in the square = stance phase right forelimb. Grey arrows indicate upward head movement, blue arrows indicate downward head movement. Upper graphs: grey lines show movements of left and right forelimb, with maximum values identifying the protracted foreleg = beginning of the stance phase (stride identification). Lower graphs: dark green line shows head movement, light green line shows movement of left forelimb; the numbers represent the frames of the video in the extracted sequence.

**Table 1 animals-12-02804-t001:** Stifle reference point—Hindlimb-lame horses.

Horse	Lameness	Degree of Lameness (1–5)		CL/CR	$DS_{St}(\mathrm{CL})$ $DS_{St}(\mathrm{CR})$	$\mathrm{Difference}\;D_{St}\;$ $=|DS_{St}$ $\left(\mathrm{CL})-\;DS_{St}(\mathrm{CR})\right|$	Classified Lame Based on AI
		1–2	3–4				
1	LH		X	CL CR	42.50 44.17	1.67	No
2	RH		X	CL CR	42.17 34.32	7.85	Yes
3	RH	X		CL CR	31.16 29.69	1.47	Yes
4	LH		X	CL CR	47.68 54.61	6.93	Yes
5	RH	X		CL CR	43.32 42.09	1.23	Yes
6	LH	X		CL CR	36.20 38.21	2.01	Yes
7	LH		X	CL CR	48.03 51.12	3.09	Yes
8	LH	X		CL CR	47.36 49.60	2.24	Yes
9	RH		X	CL CR	49.90 38.55	11.35	Yes

RH = Right hindlimb, LH = Left hindlimb, CL = Circle left, CR = Circle right.

**Table 2 animals-12-02804-t002:** Stifle reference point—Non-lame horses.

Horse	CL/CR	$DS_{St}(\mathrm{CL})$ $DS_{St}(\mathrm{CR})$	$\mathrm{Difference}\;D_{St}\;$ $=|DS_{St}$ $\left(\mathrm{CL})-\;DS_{St}(\mathrm{CR})\right|$	Classified Sound Based on AI
1	CL CR	38.27 37.76	0.51	Yes
2	CL CR	35.82 34.95	0.87	Yes
3	CL CR	40.44 39.75	0.69	Yes
4	CL CR	46.58 46.51	0.07	Yes
5	CL CR	46.09 45.93	0.16	Yes
6	CL CR	42.35 41.53	0.82	Yes
7	CL CR	37.43 36.19	1.24	No
8	CL CR	40.18 40.18	0.	Yes

**Table 3 animals-12-02804-t003:** Tuber coxae reference point—Hindlimb-lame horses.

Horse	Lameness	Degree of Lameness (1–5)		CL/CR	$DS_{Tcox}(\mathrm{CL})$ $DS_{Tcox}(\mathrm{CR})$	$\mathrm{Difference}\;D_{Tcox}\;$ $=|DT_{Tcox}$ $\left(\mathrm{CL})-\;DT_{Tcox}(\mathrm{CR})\right|$	Classified Lame Based on AI
		1–2	3–4				
1	LH		X	CL CR	11.29 19.21	7.92	No
2	RH		X	CL CR	13.18 12.17	1.01	No
3	RH	X		CL CR	11.81 14.62	2.81	Yes
4	LH		X	CL CR	15.68 20.89	5.21	No
5	RH	X		CL CR	9.22 9.95	0.73	Yes
6	LH	X		CL CR	11.53 12.13	0.60	No
7	LH		X	CL CR	13.69 15.02	1.33	No
8	LH	X		CL CR	7.98 10.36	2.38	No
9	RH		X	CL CR	11.18 11.27	0.09	Yes

**Table 4 animals-12-02804-t004:** Tuber coxae reference point—Non-lame horses.

Horse	CL/CR	$DS_{Tcox}(\mathrm{CL})$ $DS_{Tcox}(\mathrm{CR})$	Difference $D_{Tcox}\;$$=|DT_{Tcox}$$\left(\mathrm{CL})-\;DT_{Tcox}(\mathrm{CR})\right|$	Classified Sound Based on AI
1	CL CR	11.13 11.82	0.69	Yes
2	CL CR	12.06 11.55	0.51	Yes
3	CL CR	14.28 19.06	4.78	No
4	CL CR	13.99 14.49	0.50	Yes
5	CL CR	11.38 11.81	0.43	Yes
6	CL CR	9.96 10.64	0.68	Yes
7	CL CR	8.45 9.59	1.14	No
8	CL CR	8.15 9.79	1.64	No

**Table 5 animals-12-02804-t005:** Diagnostic test characteristics SE, SP, ACC, PPV and NPV of forelimb and hindlimb classification based on AI calculations when compared to the full clinical assessment (reference) in a study of 22 horses with lameness and eight horses without lameness (calculations of table contents based on Table A3, Table A4, Table A5 and Table A6)).

Test	True Positive	False Positive	False Negative	True Negative	SE (%)	SP (%)	AC (%)	PPV (%)	NPV (%)
**Forelimb AI**	13	0	0	8	100	100	100	100	100
**Hindlimb AI stifle**	8	1	1	7	88.9	87.5	88.2	88.9	87.5
**Hindlimb AI** **tuber coxae**	3	3	6	5	33.3	62.5	47.1	50	45.4

## Data Availability

The data presented in this study are available on request from the corresponding author.

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
