# Peer review of "Artificial Intelligence for Lameness Detection in Horses—A Preliminary Study"

_animals, 2022, doi:10.3390/ani12202804_

Round 1

Reviewer 1 Report

I think that the paper presents an interesting and innovating point of view in the horse lameness field and poses the bases for more specific applications and development.

However, I think the authors should specify that one of the limitations of this system is the lack of anatomical localization of the lameness site. I agree with the authors that, in subtle lameness, is quite difficult to recognize and that, often, horses referred for poor performance are lame. 

It would be interesting to evaluate the same horse prior and after diagnostic blocks.

In my opinion, the described tool can be more useful for training of both student and clinicians and in case of subtle gait changes. 

All the most clinical parts of the manuscript need to be improved due to some inaccuracies and to be edited by a mother language. 

Line 25: Change 58 in Fifty-eight; it is not correct to start sentences with acronyms or numbers

Line 30: the dot is missing

Line 30-31: I think that data concerning Stifle point of t.coxae point are pointless and confusing here

Line 185: You evaluated horses during walk and trot. I think that is ethically incorrect trotting an horse with a 4 grade lameness and I have some concern also about horse with a 3 grade lameness. If these horses were evaluated only on walk, please specify.  

194: Computed tomography, not computer tomography

309: Results: do you consider the different location of the lameness? Are there some differences in the graphical presentation between a horse with a foot lameness and another one with hock lameness? 

388: I disagree. You don’t need expensive equipment to evaluate a lame horse, you just need two eyes… please rephrase.

399: Head nodding can be seen also in horses with severe hindlimb lameness, do you consider this bias? Do you have false positive forelimb horse?

Author Response

Dear reviewer,

thank you for looking at our manuscript and providing us with very valuable and highly appreciated input. Wee tried to change the manuscript accordingly.

Best regards, Anna May

Below you find our answers to your questions/points: 

I think that the paper presents an interesting and innovating point of view in the horse lameness field and poses the bases for more specific applications and development.

However, I think the authors should specify that one of the limitations of this system is the lack of anatomical localization of the lameness site.

-Information has been added in L 488.

I agree with the authors that, in subtle lameness, is quite difficult to recognize and that, often, horses referred for poor performance are lame. 

It would be interesting to evaluate the same horse prior and after diagnostic blocks.

This is very interesting indeed. Further studies are planned for this kind of evaluation.

In my opinion, the described tool can be more useful for training of both student and clinicians and in case of subtle gait changes. 

That’s correct. We highlighted this in line 50 and in the discussion. Would you like us to emphasize more on the matter?

All the most clinical parts of the manuscript need to be improved due to some inaccuracies and to be edited by a mother language. 

The clinical parts have been edited by two independent native speakers.

Line 25: Change 58 in Fifty-eight; it is not correct to start sentences with acronyms or numbers

This has been changed (L 24)

Line 30: the dot is missing

This has been changed.

Line 30-31: I think that data concerning Stifle point of t.coxae point are pointless and confusing here

The sentence has been rephrased (L 29).

Line 185: You evaluated horses during walk and trot. I think that is ethically incorrect trotting an horse with a 4 grade lameness and I have some concern also about horse with a 3 grade lameness. If these horses were evaluated only on walk, please specify.  

Due to ethical concerns horses with lameness grade 4 and 5 were excluded; horses showing grade 3 were trotted.

194: Computed tomography, not computer tomography

This has been changed in the text (L 201).

309: Results: do you consider the different location of the lameness? Are there some differences in the graphical presentation between a horse with a foot lameness and another one with hock lameness? 

To differentiate between variable localizations (foot vs. e.g. hock) further studies are needed. From what we have discovered so far there might be the possibility to differentiate distal vs. proximal lameness. One advantage of this system compared to other systems is the magnitude of reference points, which offer many combinations for analysis, because different areas of the horses can be examined.

388: I disagree. You don’t need expensive equipment to evaluate a lame horse, you just need two eyes… please rephrase.

The sentence has been rephrased (L 413).

399: Head nodding can be seen also in horses with severe hindlimb lameness, do you consider this bias? Do you have false positive forelimb horse?

Yes, we considered this bias! Further studies on hindlimb lame horses and their head movements during lameness evaluation by the system will be performed once the program is improved.

Reviewer 2 Report

Overview and general recommendation:

I think this is an important proof-of-concept paper, as we would benefit if we could utilise pose estimation to detect lameness in horses. I enjoyed reading the background on kinetic and kinematic analysis techniques. I thought the methods were thorough and well-written.

I do have a few queries about the methodology and analysis of the results, which I will specify in the specific comments section below.  

Specific comments:

Suggest rewording the article title as it reads a bit clunkily. Perhaps something like “The usage of artificial intelligence for lameness detection in horses” or similar.

Abstract

Line 21-22: “the use of a horse” – suggest deleting this. I think welfare of the horse should suffice.

Line 30: “hindlimb lameness Stifle point can be calculated based on the data;” – is there perhaps an error in the wording of this sentence?

Introduction

Line 37: horse’s instead of horses’

Line 39: Just checking that “lost use” a typical term to use in this context?

Materials and methods

Lines 227 – 231: Nothing to action in terms of the manuscript, but can the authors please explain for my benefit how and why the reduction of the resolution of the input images improved training? 

Lines 288 – 291: This might not require any action in terms of changing the manuscript and might be more to do with my understanding of the concept instead of the way it is written - can I please get a clarification here what you mean by excluding the maximum 5% and minimum 5% of the recorded frames? I can’t picture this conceptually to see how this avoids false results.

Line 302: Missing an “a” in the work and

Results

Forelimb lameness – It is interesting to me that there is no distinct pattern of head movement in sound horses and that it is completely random. It would be beneficial to add a reference to literature that affirms this observation.  

Hindlimb lameness (Stifle) – Sorry if I have missed this somewhere in your methods or missing something conceptually, but is there a guideline/rule on how you are determining the threshold of lame classification based on your DSt compared to the control median, i.e. is there a difference that is acceptable for each category of degree of lameness?

Hindlimb lameness (T.coxae) – Similar query here on threshold of difference in DTcox in which you determine if a horse is sound versus lame?

As we seem to be comparing medians to draw conclusions about a horse’s lameness, I wonder if statistical techniques might be of benefit in augmenting your results? However, I defer to the editor’s opinion on this.

Author Response

Dear reviewer,

thank you for looking at our manuscript and providing us with very valuable and highly appreciated input. Wee tried to change the manuscript accordingly.

Best regards, Anna May

Below you find our answers to your questions/points: 

I think this is an important proof-of-concept paper, as we would benefit if we could utilise pose estimation to detect lameness in horses. I enjoyed reading the background on kinetic and kinematic analysis techniques. I thought the methods were thorough and well-written.

Thank you!

I do have a few queries about the methodology and analysis of the results, which I will specify in the specific comments section below.

Suggest rewording the article title as it reads a bit clunkily. Perhaps something like “The usage of artificial intelligence for lameness detection in horses” or similar.

The title has been reworded

Introduction

Line 37: horse’s instead of horses’

This has been changed (L 37).

Line 39: Just checking that “lost use” a typical term to use in this context?

The sentence has been rephrased.

Materials and methods

Lines 227 – 231: Nothing to action in terms of the manuscript, but can the authors please explain for my benefit how and why the reduction of the resolution of the input images improved training? 

The network topology by the original authors of DeepLabCut is more suited for smaller resolution images. This is because of the input size of the network. CNNs work by moving a so called filter (or you can say applying the convolution in a mathematical sense) over the image.

The higher the resolution is, the less passes the filter. A bit simplified: In very high resolution that filter might only see a few hairs of the horse at once. You can imagine taking a small magnifying glass and either looking at an A4 piece of paper or a billboard the size of a house. It is easier to identify e.g. the shape of the horse on the small paper. You get more detail on the billboard but you loose a sense for the whole image. To recognize patterns we need to see more, so by reducing the resolution we see more of the picture at once and can see patterns like the body shape of the horse.

Lines 288 – 291: This might not require any action in terms of changing the manuscript and might be more to do with my understanding of the concept instead of the way it is written - can I please get a clarification here what you mean by excluding the maximum 5% and minimum 5% of the recorded frames? I can’t picture this conceptually to see how this avoids false results.

With the AI not being perfect yet on recognizing the reference points throughout the video with 100% certainty, it happens that points are set on the wrong leg or “jump” to another horse in the background in a few consecutive frames (as mentioned in the discussion); these deviating numbers are left out by excluding the maximum and minimum 5%.

Line 302: Missing an “a” in the work and

This has been changed.

Results

Forelimb lameness – It is interesting to me that there is no distinct pattern of head movement in sound horses and that it is completely random. It would be beneficial to add a reference to literature that affirms this observation.  

Compensatory asymmetrical head movement can be shown in lame horses (“Head and trunk movement adaptions in horses with experimentally induced fore- or hindlimb lameness”, Buchner et al. 1996), whereas non-lame horses are more easily distracted so that their normal sinusoidal pattern of head movement does not show in all sections of the video (“Equine Locomotion”, editors: Back and Clayton, 2nd edition, 2013). With the horse being a flight animal, some horses easily get distracted, especially when filmed outdoors, which causes more “up-and-down” trend in the graphs.

Hindlimb lameness (Stifle) – Sorry if I have missed this somewhere in your methods or missing something conceptually, but is there a guideline/rule on how you are determining the threshold of lame classification based on your DSt compared to the control median, i.e. is there a difference that is acceptable for each category of degree of lameness?

Hindlimb lameness (T.coxae) – Similar query here on threshold of difference in DTcox in which you determine if a horse is sound versus lame?

As we seem to be comparing medians to draw conclusions about a horse’s lameness, I wonder if statistical techniques might be of benefit in augmenting your results? However, I defer to the editor’s opinion on this.

In this preliminary study the aim was to show the feasibility of the system and whether it was possible to differentiate between sound and lame horses by using the stifle as a reference point. To determine exact thresholds more horses have to be examined and statistics applied to a larger amount of data.

Reviewer 3 Report

animals-1891163

The paper is very interesting, written mostly in an easy and understandable way. There are however, some points that have to be  clarified and written with more precision. The structure of the paper should be corrected also. The Usage of AI for horse lameness detection seem promising, so the presented paper. Some technical details have to be added. The lack of frames per second do not allow to evaluate the work precisely. Because of limited amount of horses and breeds used I would suggest to add into the title –preliminary study. Material and methods have to be corrected for better clearance – groups of horses should be describe in detail in the same way (sections?), video tests done in each group as well (minutes are not given for each group). Statistical analysis should be explain in the special section of Material and methods- there are only results –  in result part, but also in discussion, or tables (appendix only). Statistical tests done – all test, correlations should be described earlier. Discussion is very interesting, but comparison with other results is lacking. Limitation part should be constructed as a separate part of the paper (not necessary as a section).

Detailed remarks:

L 27- one analysed group?

L 30 – not clear – many points, then only one named. Please correct the sentence

The paper has a lot of one sentence paragraphs (even the next sentence is connected starting with the word –however). Please avoid such paragraphs and correct it through all the manuscript.

L 84 – calibrated regularly – please be more clear

L 98 – drift errors could be explained , written more detailed

L 143 – pretrained – please explain

Figure 1 – reference points should be marked with numbers on photo (it is not clear why so many points are detected and only three investigated?) What about the others- are they useless?

L163 –definition of lame horses should be given by the first time you write about them or in brackets (see below)

L166 – I would suggest other title – or adding collection of data in investigated groups

All group of horses should be described in detail – the training one also, as much as possible. 65 is not enough. The same about various environment – please write something on it. The investigation should be repeatable. What kind of training was it?

Each group should be described in the same way – horses (breed, size, age), tests done in minutes and situations by the same scheme.

L 178 – where are the results for both sides and straight line? Is it calculated all together? How is it recognized by the video/further statistics?

L 184 – treatment before the examination/investigation? Please explain.

L 189-190 – one minute in trot and walk? How many strides did you manage to measure?

L 200 –how many horses  left?

L 209-210 – how many frames per second was filmed?

L 222-224 – not quite clear

L 224  and L 501– hyperparameters?

L 239 – 1.9% of what?

L 240 – likelihood 60%? Why 60? Any citation?

L 249 – data analyses and measurements – please write in detail in what gaits was it measured (in material you have trot and walk). Underline in titles?

L 260-262 – are these data presented somewhere? It may be useful for other researchers.

L 286 – the differences in lameness detection possibility (straight/circle) should be discuss in discussion

L 288-292 the possible influence of such data reduction should be discuss in limitation part. How could it be connected with the horse anatomy/conformation? Does it exclude any horse?

L 304 – it is not clear how the circle direction was taken into account in this analysis. It was connected with lameness detection I suppose.

L 308 –please give more detail information

L 320 – please add all statistical/mathematical analysis into separate section of method section

L 336 – all parameters, formula, statistical calculations should be presented first in the method section

L 377 – kappa – it should be first in the methods

L 480 - over weighted/fat ? would be better

L 530 – conclusions should be connected with results and discuss. You should underlined your conclusion in the text first.  

Author Response

Dear reviewer,

thank you for looking at our manuscript and providing us with very valuable and highly appreciated input. We tried to change the manuscript accordingly.

Best regards, Anna May

Below you find our answers to your questions/points:

The paper is very interesting, written mostly in an easy and understandable way. There are however, some points that have to be clarified and written with more precision. The structure of the paper should be corrected also. The Usage of AI for horse lameness detection seem promising, so the presented paper. Some technical details have to be added. The lack of frames per second do not allow to evaluate the work precisely. Because of limited amount of horses and breeds used I would suggest to add into the title –preliminary study. Material and methods have to be corrected for better clearance – groups of horses should be describe in detail in the same way (sections?), video tests done in each group as well (minutes are not given for each group).

Groups have been described in the same way now (breed, median age, sex) and time for different examinations listed (L 166f)

Statistical analysis should be explain in the special section of Material and methods- there are only results –  in result part, but also in discussion, or tables (appendix only). Statistical tests done – all test, correlations should be described earlier.

Statistical analysis section has been added (L 318-333)

Discussion is very interesting, but comparison with other results is lacking. Limitation part should be constructed as a separate part of the paper (not necessary as a section).

The comparison with other results (if applicable) has been added, limitation part is now a separate part of the paper

Detailed remarks:

L 27- one analysed group?

This has been changed.

L 30 – not clear – many points, then only one named. Please correct the sentence

The sentence has been changed.

The paper has a lot of one sentence paragraphs (even the next sentence is connected starting with the word –however). Please avoid such paragraphs and correct it through all the manuscript. 

Sections have been re-edited.

L 84 – calibrated regularly – please be more clear

The sentence has been rephrased.

L 98 – drift errors could be explained, written more detailed

Information has been added.

L 143 – pretrained – please explain 

Pretrained residual neural networks (ResNets) have been trained before on ImageNet, a database that provides images for large-scale object recognition models.

Figure 1 – reference points should be marked with numbers on photo (it is not clear why so many points are detected and only three investigated?) What about the others- are they useless?

Reference points have been marked with numbers. Further studies have to be performed to evaluate the other points. We picked the most promising and the ones that have been previously used in other studies.

L163 –definition of lame horses should be given by the first time you write about them or in brackets (see below)

The definition of lame horses has been specified in the text (L 161).

L166 – I would suggest other title – or adding collection of data in investigated groups 

The title has been changed.

All group of horses should be described in detail – the training one also, as much as possible. 65 is not enough. The same about various environment – please write something on it. The investigation should be repeatable. What kind of training was it?

Each group should be described in the same way – horses (breed, size, age), tests done in minutes and situations by the same scheme.

The section has been changed and comments have been added in the text (L166f).

L 178 – where are the results for both sides and straight line? Is it calculated all together? How is it recognized by the video/further statistics?

During the studies it was noticed that videos from the front/back/side were not suitable for the AI; Single frames could be used for training the AI, but the video with changing angles of view proved to be not suitable for this early state of development of the tool.

L 184 – treatment before the examination/investigation? Please explain. 

Treatment was initiated after diagnosis, the sentence has been rephrased. (L 191)

L 189-190 – one minute in trot and walk? How many strides did you manage to measure?

Information has been added in the text, (L 256f)

L 200 –how many horses left?

Information has been added (L 205).

L 209-210 – how many frames per second was filmed?

Information has been added (L 217).

L 222-224 – not quite clear

The default value is set to 5% of all images for evaluation during training. This is a common ratio between training and evaluation data. You have the option to increase it to e.g. 10%. 5% gave us only a small set of images for evaluation in absolute numbers. But without having >1000 labelled pictures for training it is important to use every picture for the training without reducing the ability to evaluate the results in the next step. With a growing dataset in the next step there is be the possibility to raise the default value.

L 224  and L 501– hyperparameters?

Parameters are all the values inside the neural network,, e.g the weights of the nodes in the network. Those are the values that are derived with training/learning. Mathematically adapting the model to the real world scenario that it should represent, changes the parameters until the representation satisfies the set target.

Hyperparameters are all values that influence and control this training process, e.g. the network type/topology, learning rate or the training/evaluation split. These are defined before training.

L 239 – 1.9% of what?

The sentence has been changed(L 245).

 The points in the image can be shifted on average by up to 1,9% regarding the size of the input images for the network. It´s the relative position the points in the image can be off from the actual position. Calculated by dividing the highest average error by the smaller dimension of the image to show how much is possible of discrepancy in the worst case: 8.22 pixels/432 pixels = ~1,9%

L 240 – likelihood 60%? Why 60? Any citation?

60% takes all the points where the network is sure enough with its prediction. The certainty is a variable value which can be adapted during training of a network. It was left close to the default the authors recommended and it is explained how to derive the value by examining the results during evaluation.

As it filters out very unlikely points that just distract the analysis, the aim is to see as many points as possible without having points that jump too much around. (Mathis et al. DeepLabCut: markerless pose estimation of user-defined body parts with deep learning. Nat Neurosci. 2018;21(9):1281-9.)

L 249 – data analyses and measurements – please write in detail in what gaits was it measured (in material you have trot and walk). Underline in titles?

The sentence has been changed and information added (L 254).

L 260-262 – are these data presented somewhere? It may be useful for other researchers.

Free accessibility of data is not possible at this point. It is intended when the program is improved.

L 286 – the differences in lameness detection possibility (straight/circle) should be discuss in discussion

To gather a sufficient number of steps for analysis only circle videos were analysed; information has been added (L 506).

L 288-292 the possible influence of such data reduction should be discuss in limitation part. How could it be connected with the horse anatomy/conformation? Does it exclude any horse?

Information has been added (L 531)

L 304 – it is not clear how the circle direction was taken into account in this analysis. It was connected with lameness detection I suppose.

As lameness should be evaluated on both hands, left and right circles were taken into account.

L 308 –please give more detail information

Statistical analysis section has been added (L 318-333)

L 320 – please add all statistical/mathematical analysis into separate section of method section

Mathematical calculations have been added (L 312).

Statistical analysis section has been added (see above).

L 336 – all parameters, formula, statistical calculations should be presented first in the method section

L 377 – kappa – it should be first in the methods

A statistical analysis section has been added (see above).

L 480 - over weighted/fat ? would be better 

The sentence has been rephrased (L 530).

L 530 – conclusions should be connected with results and discuss. You should underlined your conclusion in the text first.  

The discussion has been rearranged based on structure of the results and more references have been discussed.